# DIVIDE-AND-DENOISE: A GAME THEORETIC METHOD FOR FAIRLY COMPOSING DIFFUSION MODELS

## ABSTRACT

With the widespread availability of pre-trained diffusion models, there are many options for which models to use and how to use them together. Making these decisions depends highly on both the user's goals and the expertise of each model. Taking this into account, we propose coordinating models as one would a specialized workforce–through a fair yet efficient division of labor. Divide-and-Denoise uses multiple pre-trained diffusion models, each defined over the same space, to refine a noisy sample over time. At every timestep, we alternate between (i) dividing the sample into regions in a way that satisfies our game-theoretic criteria and (ii) denoising a region with the assigned model in a way that respects our alignment criteria. This leads to a new composite denoising process that evolves together with a division process. Since ground truth is typically not available for our setup, we measure how well Divide-and-Denoise coordinates a team of single-concept text-to-image diffusion models relative to a multi-concept model. On the GenEval benchmark, our method generates images that capture the strengths of each model, outperforming baselines and resolving common failures like missing objects and mismatched attributes.

## 1 INTRODUCTION

Large-scale diffusion models are changing the game for many disciplines. In robotics, models trained on expert demonstrations can act as long-horizon planners in unseen environments Xu et al. (2024); Chi et al. (2023); Ajay et al. (2023); Sun et al. (2023), while in biomedicine, models trained on protein structures can propose candidate therapeutics Jumper et al. (2021); Corso et al. (2023); Alamdari et al. (2023). These advancements build on a substantial body of work in computer vision, where diffusion models were introduced Song & Ermon (2019); Ho et al. (2020a), refined Geng et al. (2024); Peebles & Xie (2023); Nichol et al. (2021); Saharia et al. (2022); Burgert et al. (2023), and scaled Rombach et al. (2022); Sehwag et al. (2024). However, training effective diffusion models demands significant computational resources Sehwag et al. (2024), carries a large carbon footprint Sehwag et al. (2024), and often requires task-specific fine-tuning Black et al. (2023); Wallace et al. (2024); Chen et al. (2023). Given these challenges, there is a pressing need for effective model reuse, control, and composition Dhariwal & Nichol (2021); Ho & Salimans (2022).

Among the many available models, choosing which one to use is not always obvious. A collection of them may even be used together in order to generate data that no individual model could generate alone. Consider, as a running example, one model trained on images of dogs and another on those of cats. A common approach is to define a composite distribution as the product or mixture of the 'dog' and 'cat' densities Liu et al. (2022); Du et al. (2023). Other analytical operations include the harmonic mean and contrast Garipov et al. (2023), as well as logical operations such as AND Skreta et al. (2024). Although these operations permit tractable sampling, they are often too simple to preserve the characteristics of each model's distribution when there is conflict. For instance, if models are trained on images of animals appearing in the center, sampling from their product of densities typically produces incoherent, overlapping dogs and cats.

**Related Work.** Recent work has explored composing text-to-image diffusion models to improve spatial control Bar-Tal et al. (2023); Du et al. (2023). The typical strategy is to have the user segment an image into spatial regions, assign each region a text prompt (e.g., 'dog' or 'cat'), and then denoise each region with the corresponding model. Although simple to implement, these techniques rely on

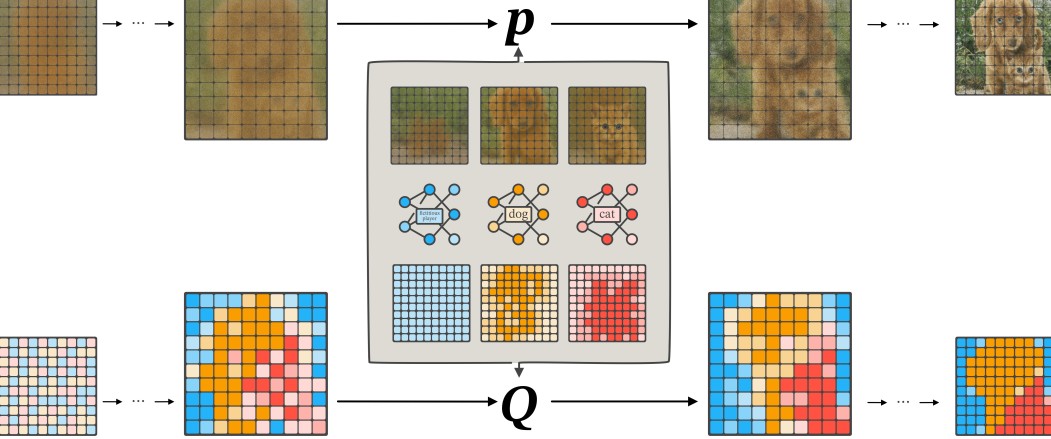

Figure 1: Divide-and-Denoise. A noisy image is iteratively refined by two coupled processes: (i) a division step that computes a fair and efficient division of the latent image given by the allocation $Q$ and (ii) a composite denoising step that reconciles the updates of several single concept text-to-image diffusion models into $p$ using this allocation. At every timestep, models provide utilities, the image is divided into soft regions in order to maximize total utility under fairness constraints, and each region is denoised with the assigned model.

user-defined allocations. This kind of division of labor between models is cumbersome to specify, infeasible to define in many domains (e.g., manually partitioning proteins), and does not take into account the strengths or weaknesses of each model. Bar-Tal et al. (2023) for example assumes these models can faithfully follow the user-prescribed layout, an assumption that often fails and requires additional forms of guidance Couairon et al. (2023); Manukyan et al. (2023). This dependence on manual segmentation restricts generalization to domains beyond computer vision.

**Contributions.** In order to address these shortcomings, we propose Divide-and-Denoise: a game-theoretic framework for coordinating multiple pre-trained diffusion models. Instead of requiring ground-truth partitions, we infer them online by requiring a division of labor among the models that is both fair and efficient. Our method is fully compositional in the sense that models need not share weights, architectures, or training data, as long as they operate in a latent space of the same dimension. We summarize our contributions below:

- We propose an inference-time algorithm for steering pre-trained diffusion models when many models with differing expertise are available.

- We introduce two processes that unfold in tandem: (i) a *division process* that we characterize as a fair division game in which each diffusion model acts as a player, and (ii) a composite *denoising process* that softly aligns each model with its assigned region.

- We find that cross-attention maps contained within diffusion models can be useful definitions of player utility. This mechanism is necessary for defining fairness and efficiency, yet general enough to allow for applications beyond vision to text-to-graph Chang & Ye (2025), text-to-audio Liu et al. (2023), and audio-to-image Biner et al. (2024) generation.

- We demonstrate our approach for leveraging the strengths of many models on the GenEval benchmark Ghosh et al. (2023). We are able to generate images from a team of single-concept text-to-image models that exceed the quality of those from a comparable multi-concept model.

## 2 BACKGROUND

### 2.1 DIFFUSION SAMPLERS

Pre-trained diffusion models with parameters $\theta$ are typically provided as time-dependent denoisers $\epsilon_t(\mathbf{x}_t; \theta)$, which predict the noise that should be removed from a sample $\mathbf{x}_t$. To generate new data from such models, we can simulate the reverse diffusion process with samplers including DDPM

Ho et al. (2020b), DDIM Song et al. (2022), and other numerical methods. All these procedures can be expressed as sampling from a sequence of Gaussian transition kernels

$$p_t(\mathbf{x}_{t-1}|\mathbf{x}_t; \theta) := \mathcal{N}(\mu_t(\mathbf{x}_t; \theta), \sigma_t^2 I), \quad t = 1, \ldots T - 1.$$

Generation begins by drawing $\mathbf{x}_T$ from the standard normal distribution, $\mathbf{x}_T \sim \mathcal{N}(0, I)$. The sampler then iteratively produces $\mathbf{x}_{T-1}, \mathbf{x}_{T-2}, \ldots, \mathbf{x}_0$ by applying these transition kernels until a final sample $\mathbf{x}_0$ is obtained. Notably, the family of samplers introduced in Song et al. (2021) allows both the total number of sampling steps $T$ and the noise schedule $\sigma_t$ to be varied while keeping the pre-trained model fixed. In this framework, the noise level is parameterized by a scalar $\eta \in [0, 1]$, where $\eta = 1$ recovers the DDPM ancestral sampler and $\eta = 0$ yields a fully deterministic trajectory. Empirical results in Song et al. (2021) show that when the number of steps $T$ is relatively small (e.g., 50–100), near-deterministic settings with $\eta$ close to zero achieve optimal performance.

## 2.2 TEXT-TO-IMAGE MODELS

Text-to-image diffusion models are an example of a conditional generative model. They are trained on massive datasets $\mathcal{D} := \{\mathbf{x}, \boldsymbol{y}\}$ where $\mathbf{x} \in \mathbb{R}^{H \times W \times 3}$ is typically a high-resolution image and $\boldsymbol{y}$ is a text prompt with many possible numerical representations. The denoising kernel $p_t$ of these models then depends on a prompt-conditioned mean $\mu_t(\mathbf{x}_t, \boldsymbol{y}; \theta)$. In practice, images are encoded into compressed representations $\phi(\mathbf{x}_t) \in \mathbb{R}^{D \times D \times 3}$ and prompts are encoded into $L$-length sequences $\tau(\boldsymbol{y}) \in \mathbb{R}^{L \times K}$.

**Attention Maps.** Modern diffusion architectures rely heavily on cross-attention layers to condition generation on the prompt Vaswani et al. (2017). A text-conditioned denoiser can be expressed as $\epsilon_t(\mathbf{x}_t, \boldsymbol{y}; \theta) = f_t(\mathbf{x}_t, \boldsymbol{y}, \{A_t(\mathbf{x}_t, \boldsymbol{y}; \theta)\}; \theta)$. Here, we make explicit the dependence of the denoiser on a set of cross-attention maps $\{A_t(\mathbf{x}_t, \boldsymbol{y}; \theta)\} \in \mathbb{R}^{\bar{D} \times \bar{D} \times K}$, one for each cross-attention layer of the network. These scores describe how much a token in the prompt attends to a given pixel. Each layer operates with a different representation of $\mathbf{x}_t$ in $\mathbb{R}^{\bar{D} \times \bar{D}}$, hence the size of these maps need not all be the same. Generation can be controlled in creative ways by substituting attention maps $\{A_t(\mathbf{x}_t, \boldsymbol{y}; \theta)\}$ with those from another pre-trained model $\{A_t(\mathbf{x}_t, \boldsymbol{y}; \theta')\}$ Hertz et al. (2022). Upscaling and averaging these maps across layers leads to a saliency map $A_t(\mathbf{x}_t, \boldsymbol{y}; \theta) \in \mathbb{R}^{D \times D}$ as shown in Tang et al. (2023).

## 2.3 FAIR DIVISION

Dividing $m$ goods among $n$ players is a classical problem in game theory Amanatidis et al. (2023); Nishimura & Sumita (2021); Dickerson et al. (2014); Cole et al. (2017); Eisenberg & Gale (1959); Caragiannis et al. (2019). In the setting of indivisible items, each player $i \in \{1, 2, \ldots, n\}$ is allocated a bundle of goods, represented by a binary assignment vector $\mathbf{M}_i \in \{0, 1\}^m$, so that no two players share any goods and all goods are allocated. Each player has a utility function $u_i : \{0, 1\}^m \to \mathbb{R}_+$ that measures the value of any bundle. Among all possible partitions, we typically seek solutions that are fair and efficient with respect to these utilities. The three main notions of fairness are: *envy-freeness* (no player prefers another player's bundle), *proportionality* (each player receives at least $1/n$ of their total utility), and *equitability* (all players receive bundles of equal utility). Efficiency can be measured, for example, by Nash social welfare, the product of individual utilities.

**Mixed Allocations.** In the case of a single good, no matter who gets it, the partition is not fair to others. This highlights that fair assignments do not always exist. One way to address this challenge is to consider randomized allocations over all possible assignments:

$$\mathbb{M}_{n,m} = \left\{ \mathbf{M} \in \{0, 1\}^{n \times m} : \sum_{i=1}^{n} \mathbf{M}_{i,j} = 1 \quad \forall 1 \leq j \leq m \right\}.$$

A mixed allocation $Q$ is a discrete distribution over $\mathbb{M}_{n,m}$. Fairness notions are defined in terms of expected utilities under $Q$. For example, an envy-free allocation $Q$ satisfies $\mathbb{E}_{\mathbf{M} \sim Q} u_i(\mathbf{M}_i) \geq \mathbb{E}_{\mathbf{M} \sim Q} u_i(\mathbf{M}_{i'})$ for all $1 \leq i \neq i' \leq n$. Note that the uniform allocation $\mathcal{U}(\mathbb{M}_{n,m})$ is always fair. Therefore, in the randomized setting, efficiency is crucial to avoid trivial solutions.

**Decomposable Allocations.** When utilities are additive in the goods, $u_i(\mathbf{M}_i) = \sum_{j=1}^m u_{ij}\mathbf{M}_{i,j}$, the expected utility of player $i$ simplifies to $\sum_{j=1}^m u_{ij}Q_{ij}$, where $Q_{ij} := \mathbb{E}_{\mathbf{M}\sim Q}\mathbf{M}_{i,j}$ is a fractional weight. We say that an allocation $Q$ is *decomposable* if

$$Q(\mathbf{M}) = \prod_{i=1}^n \prod_{j=1}^n Q_{ij}^{\mathbf{M}_{i,j}} \quad \forall \mathbf{M} \in \mathbb{M}_{n,m}.$$

Decomposable allocations are essentially equivalent to fractional allocations of $m$ divisible goods, where player $i$ receives a fraction $Q_{ij}$ of good $j$.

## 3 DIVIDE-AND-DENOISE

We study the problem of coordinating $n$ pre-trained diffusion models, each of which operates in a common latent space of dimension $m$. Models $1 \le i \le n$ may differ from one another in several ways. Without loss of generality, we assume that model $i$ is parameterized by $\theta_i$ and conditioned on a single concept represented by the prompt $\boldsymbol{y}_i$. Diffusion model $i$ defines a sequence of denoising kernels

$$p_T^i = \mathcal{N}(0, I), \quad p_t^i(\cdot \mid \mathbf{x}_t) = \mathcal{N}(\mu_t^i(\mathbf{x}_t), \sigma_t^2 I), \quad 1 \le t < T.$$

Our goal is to define a composite denoising process with kernels $p_t^c(\cdot|\mathbf{x}_t)$ that best leverages the expertise of each model. When each model is represented with a different concept or prompt $\boldsymbol{y}_i$, we expect samples from $p_t^c$ to ideally match what a single model trained on all the concepts appearing together would generate. Such a model, however, may not exist in practice for concepts that are sufficiently different.

### 3.1 SIMULATING TWO PROCESSES

The main components of our approach are outlined in Figure 1. Divide-and-Denoise generates two coupled trajectories: a sampling path of the composite denoising process, obtained by iteratively drawing $\mathbf{x}_{t-1} \sim p_t^c(\mathbf{x}_{t-1}|\mathbf{x}_t)$, and a path of the division process given by allocations $Q_t$, also obtained by iterative updates in time. We define each allocation $Q_t$ to be a distribution over $\mathbb{M}_{n,m}$, the space of partitions of the latent space of dimension $m$ across $n$ models. Since $Q_t$ specifies how the coordinates of the latent at time $t$ are distributed among the pre-trained models, it may be interpreted as a division of labor.

We initialize with $p_T^c = \mathcal{N}(0, I)$ and $Q_T = \mathcal{U}(\mathbb{M}_{n,m})$, and draw the first noisy latent as $\mathbf{x}_{T-1} \sim p_T^c$. At each of the remaining timesteps $1 \le t < T$, we update the allocation and the composite process according to the bi-level optimization:

$$Q_t = \arg\max_{Q \in \mathbb{Q}_t} \mathcal{G}_t(p_t^c, Q), \tag{1}$$

$$p_t^c(\cdot|\mathbf{x}_t) = \arg\max_{p \in \mathbb{P}_t} \mathcal{F}_t(p, Q_t). \tag{2}$$

The choice of the optimization objectives $\mathcal{G}$ and $\mathcal{F}$, and the constrained sets $\mathbb{Q}_t$ and $\mathbb{P}_t$ will be discussed in Sections 3.2 and 3.3. The goal of the first problem is to fairly and efficiently divide the latent among the individual diffusion models, while the purpose of the second problem is to choose a denoising update that best aligns with this division. The optimization objectives are expressed through a common alignment score $R_t$ with a problem-specific regularization. At the end of each timestep, we sample a denoised latent $\mathbf{x}_{t-1}$ from $p_t^c(\cdot|\mathbf{x}_t)$.

### 3.2 COMPUTING A FAIR AND EFFICIENT DIVISION

We formulate the problem of finding the next allocation (equation 1) as a fair division game, where the goods are latent coordinates and the players are the individual diffusion models. We assume that players have utilities for the goods given by $u_{ij}(\mathbf{x}, t)$, i.e. model $i$'s value for coordinate $j$ at latent $\mathbf{x}$ and timestep $t$. In the text-to-image setting, cross-attention maps have been shown to be effective indicators of the relevance of each pixel to a target word or phrase. This motivates us to define the utilities as

$$u_{ij}(\mathbf{x}, t) = \frac{A_t^j(\mathbf{x}, \boldsymbol{y}_i; \theta_i)}{\sum_{j=1}^m A_t^j(\mathbf{x}, \boldsymbol{y}_i; \theta_i)}, \tag{3}$$

where $A_t^j$ denotes the $j$-th coordinate of the attention map $A_t$. To measure efficiency of the allocation, we propose an alignment score defined over the players as

$$R_t(\mathbf{x}, Q) = \mathbb{E}_{\mathbf{M}\sim Q} \sum_{i=1}^n R_t^i(\mathbf{x}, \mathbf{M}), \quad \text{with} \quad R_t^i(\mathbf{x}, \mathbf{M}) = \sum_{j=1}^m \mathbf{M}_{i,j} \log u_{ij}(\mathbf{x}, t).$$

The objective $\mathcal{G}_t$ is given by the alignment score regularized by a Kullback-Leibler (KL) divergence term with positive weight $\beta_t > 0$:

$$\mathcal{G}_t(p, Q) = R_{t-1}(\hat{\mathbf{x}}_{t-1}(p), Q) - \beta_t D_{\mathrm{KL}}(Q \| Q_{t+1}), \tag{4}$$

where $\hat{\mathbf{x}}_{t-1}(p) = \mathbb{E}_{\mathbf{x}_{t-1}\sim p}\mathbf{x}_{t-1}$ is a lookahead prediction of the next sample. The regularization term penalizes abrupt changes between consecutive allocations, encouraging temporally smooth allocation trajectories that provide a stable signal for the composite denoising update. A hyperparameter $\beta$ controls the trade-off between efficiency and smoothness. For example, when $\beta \to \infty$ the allocation $Q_t$ remains uniform throughout generation.

A solution is constrained to lie in the set of fair allocations $\mathbb{Q}_t$. We express this constraint set as

$$\mathbb{Q}_t = \left\{ Q \in \Delta(\mathbb{M}_{n,m}) : \mathbb{E}_{\mathbf{M}\sim Q} \sum_{i=1}^n \sum_{j=1}^m \mathbf{M}_{i,j}\phi_{ij} \preceq \boldsymbol{b} \right\}, \tag{5}$$

where $\boldsymbol{b} = (b_1, \ldots, b_l)$ and $\phi_{ij} = (\phi_{ij}^1, \ldots, \phi_{ij}^l)$, for all $1 \le i \le n$ and $1 \le j \le m$, are coefficients specifying $l$ linear constraints. Despite this simple form, these sets are flexible enough to represent common notions of fairness under additive utilities. In the following example, we express fairness in the form 5.

**Example 1.** *Using a single linear inequality $\mathbb{E}_{\mathbf{M}\sim Q} \sum_{i=1}^n \sum_{j=1}^m \mathbf{M}_{i,j}\phi_{ij}^1 \preceq b_1$, we can encode the following relations:*

1. *Setting $b_1 = 0$ and $\phi_{kj}^1 = -u_{ij}(\mathbf{x}_t, t)I(k=i) + u_{ij}(\mathbf{x}_t, t)I(k=i')$ is equivalent to saying that player $i$ is not envious of player $i'$.*

2. *Setting $b_1 = 0$ and $\phi_{kj}^1 = -u_{ij}(\mathbf{x}_t, t)I(k=i) + u_{ij}(\mathbf{x}_t, t)/n$ is equivalent to constraining player $i$ to be allocated at least $1/n$ of its total utility. Alternatively, for the normalized utilities, we can set $b_1 = -1/n$ and $\phi_{kj}^1 = -u_{ij}(\mathbf{x}_t, t)I(k=i)$.*

3. *Setting $b_1 = 0$ and $\phi_{kj}^1 = -u_{ij}(\mathbf{x}_t, t)I(k=i) + u_{i'j}(\mathbf{x}_t, t)I(k=i')$ is equivalent to saying that the allocated utility of player $i$ is greater or equal to that of player $i'$.*

Clearly, by stacking inequalities, we can represent envy-free, proportional, and equitable constraints or their combinations for any number of players. It is worth noting that the uniform allocation is always fair, so the feasible set is not empty.

We conclude this section by introducing a generic solution to the optimization problem in equation 1.

**Theorem 1.** *Assume that allocation $Q_{t+1}$ is decomposable with weights $Q_{ij}^{t+1}$. Then, the optimal allocation $Q_t$ solving the fair division game 1 is also decomposable with weights*

$$Q_{ij}^t = \frac{\exp(-\langle\lambda^*, \phi_{ij}\rangle + \log u_{ij}(\hat{\mathbf{x}}_{t-1}(p), t-1)/\beta)Q_{ij}^{t+1}}{Z_j(\lambda^*)}. \tag{6}$$

*where $Z_j(\lambda^*) = \sum_{i=1}^n \exp(-\langle\lambda^*, \phi_{ij}\rangle + \log u_{ij}(\hat{\mathbf{x}}_{t-1}(p), t-1)/\beta)Q_{ij}^{t+1}$ is a normalization constant and $\lambda^*$ is a solution of a dual problem that reads*

$$\max_{\lambda \ge 0} \quad -\langle\boldsymbol{b}, \lambda\rangle - \sum_{j=1}^m \log Z_j(\lambda^*)$$

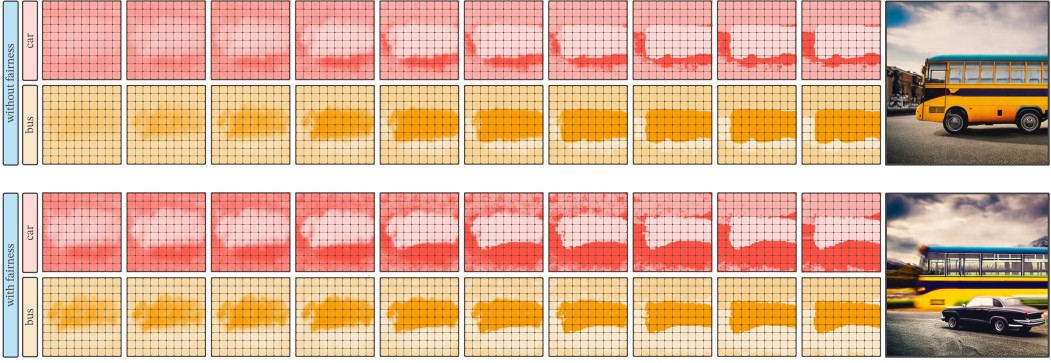

Figure 2: The role of fairness in Divide-and-Denoise. Top: without fairness, the car model is allocated significantly fewer pixels than the bus model, resulting in a missing object. Bottom: with fairness, the allocation balances both models so the car is no longer envious of the bus. Fairness prevents a single model from dominating.

## 3.3 ALIGNING MODELS WITH THEIR ALLOCATIONS

In this section, we address the second problem (equation 2) of selecting a composite denoising kernel $p_t^c$ from the set of feasible denoising distributions $\mathbb{P}_t$. We introduce a new objective that explicitly aligns each diffusion model's proposal with its assigned region, conditioned on the given allocation $Q$. Let $p_j$ denote the marginal of a denoising kernel $p$ at coordinate $j$. We define

$$\mathcal{F}_t(p, Q) = \mathbb{E}_{\mathbf{x}_{t-1} \sim p} R_{t-1}(\mathbf{x}_{t-1}, Q) - \alpha_t \mathbb{E}_{\mathbf{M} \sim Q} \left[ \sum_{i=1}^{n} \sum_{j=1}^{m} \mathbf{M}_{i,j} D_{\mathrm{KL}}(p_j(\cdot|\mathbf{x}_t) || p_j^i(\cdot|\mathbf{x}_t)) \right] \quad (7)$$

The aim of the KL regularization is to keep the denoising update for the coordinates allocated to model $i$ close to its proposal. A hyperparameter $\alpha_t > 0$ controls the trade-off between alignment with the given allocation and adherence to the proposals of individual models. Notice how each player's utility $u_{ij}(\mathbf{x}_{t-1}, t-1)$ contributes to the alignment score $R_{t-1}^i$ only if $\mathbf{M}$ assigns pixel $j$ to model $i$. Maximizing $R_{t-1}^i$ over $\mathbf{x}$ therefore encourages each model to exhibit high attention in its allocated region and low attention elsewhere. See Appendix A.1 for the policy learning interpretation of this objective.

Because the fair division game 1 partitions all $m$ goods among $n$ players, a model $i$ can be allocated a region that contains coordinates $j$ with low utility $u_{ij}$. As a result, our alignment score would encourage each model to spread its attention across a region that is larger than its region of interest. In order to prevent artifacts from a model denoising areas of little interest, we introduce a fictitious $(n+1)$-st player. This player is given a fixed uniform utility and its denoising kernel is defined as the geometric mean of the individual model kernels:

$$p^{n+1}(\mathbf{x}_{t-1}|\mathbf{x}_t) \propto \prod_{i=1}^{n} p^i(\mathbf{x}_{t-1}|\mathbf{x}_t)^{1/n}, \quad u_{(n+1)j} \equiv 1/m \quad \forall 1 \le j \le m.$$

Our fictitious player then models regions that are not relevant to any diffusion model. The allocation $Q_t$ is now extended to a distribution over $\mathbb{M}_{n+1,m}$. Since the objective 2 can be non-linear, we cannot find an explicit solution in general. However, a solution exists under simplifying assumptions.

**Theorem 2.** *Consider the optimization in equation 2 with the following assumptions: (1) $\mathbb{P}_t$ is a set of all distributions on $\mathbb{R}^m$ with independent coordinates and (2) $R_t$ is linear jointly in the first argument and $t$. For each $i \le n$, define the marginal weight vector $\hat{Q}_i$ as $\mathbb{E}_{\mathbf{M} \sim Q_t} \mathbf{M}_i$ if the fictitious player is not used, and as $\mathbb{E}_{\mathbf{M} \sim Q_t}(\mathbf{M}_i + \mathbf{M}_{n+1}/n)$ otherwise.*

*Then, the optimal composite denoising kernel is given by $p_t^c(\cdot|\mathbf{x}_t) = \mathcal{N}(\mu_t, \sigma_t^2 I)$, where*

$$\mu_t = \sum_{i=1}^{n} \mu_t^i(\mathbf{x}_t) \odot \hat{Q}_i + \frac{\sigma_t^2}{\alpha_t} \nabla_{\mathbf{x}_t} R_t(\mathbf{x}_t, Q).$$

Remarkably, the solution naturally decomposes into two parts: a compositional update given by the first term and a guidance update given by the gradient term. Furthermore, when the influence of the

---

**Algorithm 1** Coordinated Denoising with Multiple Pre-Trained Diffusion Models

---

1: **Input:** Pre-trained diffusion denoisers $\{\epsilon_t^i(\cdot)\}_{i=1}^n$ conditioned on prompts $\{y_i\}_{i=1}^n$, hyperparameters $\alpha > 0$, $\beta > 0$.

2: **function** FORWARD($\mathbf{x}, t$)
3:     **for** each denoiser $\epsilon_t^i$ **do**
4:         Compute denoised mean $\mu_t^i(\mathbf{x})$
5:         Aggregate attention scores $\{A_t^j(\mathbf{x}, \boldsymbol{y}_i)\}_{j=1}^m$
6:         Compute utilities $u_{ij}(\mathbf{x}, t) = A_t^j(\mathbf{x}, \boldsymbol{y}_i)/\sum_{k=1}^m A_t^k(\mathbf{x}, \boldsymbol{y}_i)$
7:     **end for**
8:     **return** $\{\mu_t^i(\mathbf{x})\}_{i=1}^n$, $\{u_{ij}(\mathbf{x}, t)\}_{i=1, j=1}^{n,m}$
9: **end function**

10: Initialize coordinated denoising process $p_T^c = \mathcal{N}(0, I)$ and allocation $Q_T = \mathcal{U}(\mathcal{M}_{n,m})$.
11: **for** $t = T - 1, \ldots, 0$ **do**
12:     Sample $\mathbf{x}_t \sim p_{t+1}^c$
13:     $\{\mu_t^i(\mathbf{x}_t)\}$, $\{u_{ij}(\mathbf{x}_t, t)\} \leftarrow$ FORWARD($\mathbf{x}_t, t$)
14:     Compute lookahead prediction $\hat{\mathbf{x}}_{t-1} = \sum_{i=1}^n \mu_t^i(\mathbf{x}_t) \odot \hat{Q}_i^{t+1}$
15:     $\_$, $\{u_{ij}(\hat{\mathbf{x}}_{t-1}, t - 1)\} \leftarrow$ FORWARD($\hat{\mathbf{x}}_{t-1}, t - 1$)
16:     Define fairness coefficients $\phi_{ij}$ from utilities $\{u_{ij}(\mathbf{x}_t, t)\}$
17:     Solve dual problem to obtain $\lambda^*$
18:     Update allocations $Q_{ij}^t \propto \exp(-\langle \lambda^*, \phi_{ij}\rangle + \frac{1}{\beta} \log u_{ij}(\hat{\mathbf{x}}_{t-1}, t - 1))Q_{ij}^{t+1}$
19:     Update denoising kernel $p_t^c = \mathcal{N}(\sum_{i=1}^n \mu_t^i(\mathbf{x}_t) \odot \hat{Q}_i^t + \sigma_t \frac{\nabla_{\mathbf{x}_t} R_t(\mathbf{x}_t, Q)}{\alpha \|\nabla_{\mathbf{x}_t} R_t(\mathbf{x}_t, Q)\|}, \sigma_t^2 I)$
20: **end for**

---

regularization increases ($\alpha_t \rightarrow \infty$), the optimal solution converges to the standard MultiDiffusion (Bar-Tal et al. (2023)) update.

In practice, we propose to use a local linearization technique. By applying a first-order Taylor expansion to linearize the reward, we obtain the following approximation:

$$p_t^c(\cdot|\mathbf{x}_t) \approx \mathcal{N}\left(\sum_{i=1}^n \mu_t^i(\mathbf{x}_t) \odot \hat{Q}_i + \frac{\sigma_t^2}{\alpha_t}\sum_{i=1}^n \sum_{i=j}^m Q_{ij}\nabla_{\mathbf{x}_t} \log u_{ij}(\mathbf{x}_t, t), \sigma_t^2 I\right). \tag{8}$$

We observe that performance is sensitive to the hyperparameter $\alpha_t$. Large values of $\alpha_t$ suppress the influence of the guidance term, while overly small values may lead to out-of-distribution samples. We find it useful to reparametrize $\alpha_t$ as

$$\alpha_t = \alpha\sigma_t \|\nabla_{\mathbf{x}_t} R_t(\mathbf{x}_t, Q)\|,$$

where $\alpha$ is a constant independent of time.

Assuming timesteps are sufficiently small, so that both the noisy latents and their corresponding attention maps change only minimally, we can treat the optimal allocation from the previous step as a good approximation for the optimal allocation at the current step. Consequently, we employ a warm-start approximation of the composite denoising update $p_t^c$, given by

$$\mathcal{N}\left(\sum_{i=1}^n \sum_{i=1}^n \mu_t^i(\mathbf{x}_t) \odot \hat{Q}_i^{t+1}, \sigma_t^2 I\right)$$

and perform only a single update on $Q_t$ and $p_t^c$. See Algorithm 1 for details.

## 4 EXPERIMENTS

Evaluating the effectiveness of coordinating pre-trained diffusion models with Divide-and-Denoise is challenging due to the lack of ground truth for our problem. It is not obvious how to divide work among a team of models, especially when their relative strengths and how they relate to one another

is not known. We simulate these practical challenges by choosing each model to be a text-to-image model conditioned on a single-concept text prompt. We argue that the following baselines are the most representative for our setting:

**Implied Allocation.** We construct a single, multi-concept model by conditioning a text-to-image model on a multi-concept joint prompt. This represents a baseline where the division of labor is implied by default through the output of a single model. We intentionally avoid enriching the joint prompt with information beyond the concept set $\mathcal{Y}$, such as relationships between concepts $\boldsymbol{y}_1, \boldsymbol{y}_2 \in \mathcal{Y}$, since a team of specialized models would not typically have access to this information. For model $i$ conditioned on a single concept $\boldsymbol{y}_i$, we use the prompt "an image with $\boldsymbol{y}_i$". A joint prompt is then constructed as "an image with $\boldsymbol{y}_1$ and $\boldsymbol{y}_2$ and ... and $\boldsymbol{y}_n$".

**Uniform Allocation.** We construct a composite denoising process as the mean of the proposals of each single-concept text-to-image model. This is a popular approach to compositional sampling (Liu et al., 2022). Note that this represents a baseline where the division of labor among models is uniform. Our algorithm reduces to the uniform allocation in the limit $\alpha, \beta \to \infty$.

We use the Stable Diffusion 2.0 model (Rombach et al., 2022) with a DDIM scheduler (Song et al., 2021), setting $T = 50$ sampling steps and a noise scale of $\eta = 0.01$. If not specified otherwise, all experiments with Divide-and-Denoise use hyperparameters $\alpha = 1$ and $\beta = 1$ in equation 4 and equation 7.

## 4.1 EVALUATION METRICS

We evaluate our method along three axes: how well we can generate images that contain more than one concept, how these images look when concepts become more complex, and whether images can be produced through coordinating models that are intentionally represented with conflicting concepts. In order to quantify these criteria, we make use of a popular benchmark for text-to-image generation called GenEval (Ghosh et al., 2023). GenEval leverages pre-trained segmentation and classification models trained on the COCO dataset (Lin et al., 2014) to automatically verify the presence, count, color, and spatial arrangement of objects in the generated images. These automated checks are strongly correlated with human judgments, making GenEval a reliable and scalable proxy for human evaluation. We verify whether an image contains a concept from $\mathcal{Y}$ using GenEval. We report two specific metrics:

- **% images**: The fraction of generated images that contain all concepts.
- **% prompts**: The fraction of prompts where at least one generated image contains all concepts. For each prompt 4 seeds are used.

We complement GenEval with three widely-used performance metrics, each defined as $s_i(\mathbf{x}_0, \boldsymbol{y})$. CLIP-Score ($s_1$) measures text–image alignment by computing the cosine similarity between CLIP embeddings of the generated image $\mathbf{x}_0$ and a prompt $\boldsymbol{y}$ (Radford et al., 2021). Higher scores indicate stronger global alignment. ImageReward ($s_2$) uses a learned reward model trained on large-scale human preference data to assign a single score reflecting how closely image $\mathbf{x}_0$ matches a text prompt $\boldsymbol{y}$ (Xu et al., 2023). Finally, we use a Visual Question-Answering (VQA) system called VQAScore ($s_3$) to assess how faithful $\mathbf{x}_0$ is to a prompt $\boldsymbol{y}$ (Lin et al., 2024). VQA systems assign a score by first generating questions based on the prompt $\boldsymbol{y}$, and then answering them by inspecting the generated image $\mathbf{x}_0$. For each of these three scores, we report two metrics:

- **joint**: the mean score $s_i(\mathbf{x}_0, \boldsymbol{y})$ over each generated image $\mathbf{x}_0$ and multi-concept prompt containing $\boldsymbol{y}_1, \boldsymbol{y}_2, ..., \boldsymbol{y}_n$.
- **min**: the mean score $\min_{\boldsymbol{y} \in \mathcal{Y}} s_i(\mathbf{x}_0, \boldsymbol{y})$ over each generated image $\mathbf{x}_0$ and set of single-concept prompts $\mathcal{Y}$.

## 4.2 CONCEPTS AS OBJECTS

We first evaluate our method along the axis of generating more than one concept. In this setting, we define each concept as a distinct object. We assess how well our method works for generating images with multiple objects by randomly sampling $n$ objects from the COCO vocabulary and defining a

Figure 3: On various compositional tasks, Divide-and-Denoise avoids object overlap, preserves all objects, and correctly attributes colors, outperforming baselines.

| # | Allocation | GenEval ↑ | | CLIP ↑ | | ImageReward ↑ | | VQA ↑ | |
| | | %images | %prompt | joint | min | joint | min | joint | min |
|---|---|---|---|---|---|---|---|---|---|
| | Implied | 54.85% | 84.69% | 27.06 | 18.76 | 0.31 | -1.14 | 0.758 | 0.684 |
| | Uniform | 31.63% | 63.27% | 26.43 | 18.83 | -0.40 | -1.40 | 0.744 | 0.639 |
| 2 | Efficient | **85.97%** | 98.98% | 29.60 | 21.17 | 1.12 | -0.49 | 0.958 | 0.924 |
| | Efficient + Envy-free | 85.71% | 98.98% | 29.64 | 21.23 | 1.13 | **-0.47** | **0.960** | **0.925** |
| | Efficient + Equitable | **85.97%** | 98.98% | 29.55 | 21.24 | 1.12 | -0.48 | 0.959 | 0.920 |
| | Efficient + Proportional | 84.95% | 98.98% | **29.66** | **21.24** | **1.15** | **-0.47** | 0.959 | 0.919 |
| | Implied | 14.00% | 39.00% | 28.45 | 15.25 | -0.16 | -1.82 | 0.528 | 0.386 |
| | Uniform | 3.00% | 11.00% | 25.57 | 16.14 | -1.24 | -2.04 | -1.245 | -2.038 |
| 3 | Efficient | 52.75% | 87.00% | 32.33 | 18.25 | 0.97 | -0.99 | 0.862 | 0.753 |
| | Efficient + Envy-free | 51.50% | 88.00% | 32.62 | 18.57 | 1.07 | -0.93 | 0.898 | 0.794 |
| | Efficient + Equitable | 55.00% | 88.00% | 32.49 | 18.80 | 1.05 | -0.96 | 0.907 | **0.816** |
| | Efficient + Proportional | **55.25%** | 89.00% | **32.85** | **18.83** | **1.14** | **-0.86** | **0.910** | 0.812 |

Table 1: Performance of Divide-and-Denoise in coordinating 2 and 3 models conditioned on objects as concepts.

single concept prompt based on each object. In total, we construct $98$ unique $n$-object tuples and evaluate each tuple across $4$ different seeds. Results are presented in Table 1 where row 2 and 3 correspond to $n = 2$ and $n = 3$ object-specific models.

We find that improvement over baselines is driven by the efficient division of labor. Fairness improves most metrics further. Observe that the importance of fairness increases as more models become involved, since the probability of a model being neglected by an efficient (but not fair) allocation grows. To illustrate the effect of the fairness constraint, we provide a qualitative example in Figure 2.

### 4.3 CONCEPTS WITH DESCRIPTION

We next evaluate how our method compares to baselines when concepts contain greater detail. This simulates a scenario where a single multi-concept model would typically fail. We attach color descriptions to objects, testing whether our method can faithfully bind attributes to the correct objects. Prompts are constructed as earlier, but this time with each concept $y \in \mathcal{Y}$ given by a color and object, e.g. "an image with a *red car*". We generate 2 object–color descriptions to define a pair of specialized models. In total, we construct $100$ unique pairs and evaluate each pair across $4$ different seeds. Results are presented in Figure 2, and Figure 3 provides a qualitative example of correct attribute binding.

### 4.4 CONCEPTS WITH CONFLICT

Finally, we evaluate how well Divide-and-Denoise coordinates models with conflicting interests. In order to simulate this, we hand-design 40 scenarios where concepts among the models would naturally conflict. For example, in the two model setup we condition the first model on the concept "desert" while the second on the concept "snowy mountain". A full list of single-concept prompts is provided in the Appendix. We compare Divide-and-Denoise with an efficient and envy-free allocation to the implied and uniform baselines. As shown in Table 3, our method outperforms both the uncoordinated and single model baselines.

| Allocation | GenEval ↑ | | CLIP ↑ | | ImageReward ↑ | | VQA ↑ | |
| --- | --- | --- | --- | --- | --- | --- | --- | --- |
| | %images | %prompt | joint | min | joint | min | joint | min |
| Implied | 13.75% | 36.00% | 29.77 | 19.56 | 0.13 | -1.48 | 0.593 | 0.449 |
| Uniform | 8.25% | 21.00% | 28.55 | 19.88 | -0.39 | -1.64 | 0.623 | 0.485 |
| Efficient+Fair | **45.75%** | **75.00%** | **32.13** | **22.25** | **1.10** | **-0.74** | **0.856** | **0.762** |

Table 2: Performance of Divide-and-Denoise on coordinating models conditioned on descriptive concepts.

| Allocation | CLIP ↑ | | ImageReward ↑ | | VQA ↑ | |
| --- | --- | --- | --- | --- | --- | --- |
| | joint | min | joint | min | joint | min |
| Implied | 29.68 | 19.73 | 0.63 | -0.91 | 0.747 | 0.621 |
| Uniform | 27.79 | 19.12 | -0.34 | -1.41 | 0.692 | 0.526 |
| Efficient+Fair | **30.98** | **20.94** | **1.10** | **-0.56** | **0.900** | **0.802** |

Table 3: Performance of Divide-and-Denoise on coordinating models with conflicting interests.

## 5 CONCLUSION

In this work, we introduced Divide-and-Denoise, a game-theoretic framework for coordinating several pre-trained diffusion models. Our coupled division and denoising processes resolve conflicts between models, prevent concepts or models from being neglected, and outperforms baselines across a broad range of metrics on GenEval. Most importantly, our formulation is not tied to the image domain: any generative model that provides token or region-level importance scores can in principle serve as a player with a well-defined utility function. We hope to extend our method to text-to-audio, text-to-graph, or other multi-modal synthesis settings where explicit spatial layouts are unavailable. An open question remains on how to define utilities in non-visual domains where attention maps may not be directly interpretable. Our results nevertheless highlight cooperative interaction between pre-trained models as a general recipe for controllable and reusable generative modeling across domains.

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

# A APPENDIX

## A.1 POLICY LEARNING PERSPECTIVE

To motivate compositional sampling objective equation 7, we reinterpret the compositional generation problem as a one-step Markov Decision Process (MDP).

Let states in our MDP coincide with latent representations. One denoising step corresponds to predicting the denoised latent $\mathbf{x}_{t-1}$ given the current state $\mathbf{x}_t$. An action in this MDP is the selection of $\mathbf{x}_{t-1}$, and the transition dynamic is an identity map, so the final state coincides with the action. The composite kernel $p$ plays the role of a stochastic policy, specifying the probability of each action given the current state. Pretrained diffusion models define reference policies: for an assignment matrix $\mathbf{M}$ the reference policy for the region $\{j : \mathbf{M}_{i,j} = 1\}$ is given by $p^i(\cdot|\mathbf{x}_t)$. Our goal is to approximate an optimal policy that maximizes the alignment reward $R_{t-1}(\mathbf{x}_{t-1}, Q)$ for the final state $\mathbf{x}_{t-1}$, while remaining close to the reference policies.

## A.2 PROOFS OF THE THEOREMS

In our proofs, we will rely on the following technical fact:

**Lemma 1.** *Let $\pi$ and $\pi'$ be probability distributions on the same domain. The solution $\pi^*$ to an unconstrained optimization problem*

$$\max_{\pi} \mathbb{E}_{z \in \pi} f(z) - \gamma D_{\mathrm{KL}}(\pi||\pi')$$

*is given by*

$$\pi^*(z) = \frac{\exp(f(z)/\gamma)\pi'(z)}{\int \exp(f(z)/\gamma)\pi'(z)dz}.$$

*Proof.* It is sufficient to notice that

$$\gamma D_{\mathrm{KL}}(\pi||\pi^*) = -\mathbb{E}_{z \in \pi} f(z) + \gamma D_{\mathrm{KL}}(\pi||\pi') + C,$$

where $C$ is a constant that does not depend on $\pi$. □

*Proof of Theorem 2.* Let us first consider the setting without fictitious player.

Recall that denoising kernels of individual models are Gaussians with the same covariance:

$$p^i(\cdot \mid \mathbf{x}_t) = \mathcal{N}(\mu_t^i(\mathbf{x}_t), \sigma_t^2 I).$$

Denote the marginal distribution of $p^i$ at the coordinate $j$ as

$$p_j^i(\cdot \mid \mathbf{x}_t) = \mathcal{N}(\mu_j^i(\mathbf{x}_t), \sigma_t^2).$$

For an allocation $Q$ with weights $Q_{ij}$, consider a distribution $p_t^Q$ defined as

$$p_t^Q(\mathbf{x}_{t-1}|\mathbf{x}_t) \propto \prod_{i=1}^n \prod_{j=1}^m p_j^i(\mathbf{x}_{t-1}|\mathbf{x}_t)^{Q_{ij}}.$$

Notice that

$$
\begin{aligned}
p^Q(\mathbf{x}|\mathbf{x}_t) &\propto \prod_{i=1}^n \prod_{j=1}^m p_j^i(\mathbf{x}|\mathbf{x}_t)^{Q_{ij}} \\
&= \prod_{i=1}^n \prod_{j=1}^m \exp\left( \frac{-Q_{ij}(\mathbf{x}_j - \mu_j^i(\mathbf{x}_t))^2}{2\sigma_t^2} \right) \\
&= \exp\left( \frac{-\sum_{i=1}^n \sum_{j=1}^m Q_{ij}(\mathbf{x}_j - \mu_j^i(\mathbf{x}_t))^2}{2\sigma_t^2} \right) \\
&\propto \exp\left( \frac{-\sum_{i=1}^n \sum_{j=1}^m [Q_{ij}\mathbf{x}_j^2 - 2\mathbf{x}_j \mu_j^i(\mathbf{x}_t)]}{2\sigma_t^2} \right) \\
&= \exp\left( \frac{-\sum_{j=1}^m \mathbf{x}_j^2 + 2\sum_{j=1}^m \langle \mathbf{x}_j, \sum_{i=1}^n Q_{ij}\mu_j^i(\mathbf{x}_t)\rangle}{2\sigma_t^2} \right) \\
&= \exp\left( \frac{-\|\mathbf{x}\|^2 + 2\langle \mathbf{x}, \sum_{i=1}^n Q_i \odot \mu^i(\mathbf{x}_t)\rangle}{2\sigma_t^2} \right),
\end{aligned}
$$

and thus,

$$p_t^Q(\cdot|\mathbf{x}_t) = \mathcal{N}\left( \mu_t^Q(\mathbf{x}_t), \sigma_t^2 I \right), \quad \mu_t^Q(\mathbf{x}_t) = \sum_{i=1}^n Q_i \odot \mu_t^i(\mathbf{x}_t).$$

For any distribution $p \in \mathbb{P}_t$, we have

$$p(\mathbf{x}) = \prod_{j=1}^m p_j(\mathbf{x}).$$

Therefore, the following equality holds

$$
\begin{aligned}
\mathbb{E}_{\mathbf{M}\in Q} \sum_{i=1}^n \sum_{j=1}^m \mathbf{M}_{i,j} D_{\mathrm{KL}}(p_j(\cdot)||p_j^i(\cdot|\mathbf{x}_t)) &= \sum_{i=1}^n \sum_{j=1}^m Q_{ij} D_{\mathrm{KL}}(p_j(\cdot)||p_j^i(\cdot|\mathbf{x}_t)) \\
&= \sum_{j=1}^m \left[ \sum_{i=1}^n Q_{ij} \int p_j(\mathbf{x}_j) \log p_j^i(\mathbf{x}_j|\mathbf{x}_t)d\mathbf{x}_j - \sum_{i=1}^n Q_{ij} H(p_j) \right] \\
&= \int p(\mathbf{x}) \log \prod_{i=1}^n \prod_{j=1}^m p_j^i(\mathbf{x}_j|\mathbf{x}_t)^{Q_{ij}} d\mathbf{x} - \sum_{j=1}^m H(p_j) \\
&= \int p(\mathbf{x}) \log p^Q(\mathbf{x}|\mathbf{x}_t)d\mathbf{x} - H(p) + C \\
&= D_{\mathrm{KL}}(p||p_t^Q(|\mathbf{x}_t)) + C,
\end{aligned}
$$

where $C$ is a constant that does not depend on $p$.

By Lemma 1, the composite kernel $p_t^c$ maximizing $\mathcal{F}_t(\mathbf{x}_t, p, Q)$ is given by

$$p_t^c(\mathbf{x}_{t-1}|\mathbf{x}_t) \propto \exp(R_{t-1}(\mathbf{x}_{t-1}, Q)/\alpha_t)p_t^Q(\mathbf{x}_{t-1}|\mathbf{x}_t). \tag{9}$$

Since $R_t(\mathbf{x}, Q)$ is linear jointly in $t$ and $\mathbf{x}$, we have

$$R_{t-1}(\mathbf{x}_{t-1}, Q) = R_t(\mathbf{x}_t, Q) + A(\mathbf{x}_{t-1} - \mathbf{x}_t) - b,$$

where $A = \nabla_{\mathbf{x}} R_t(\mathbf{x}_t, Q)$ and $b = \nabla_t R_t(\mathbf{x}_t, Q)$.

Substituting in equation 9, we obtain

$$
\begin{aligned}
p_t^c(\mathbf{x}_{t-1}|\mathbf{x}_t) &\propto \exp(R_{t-1}(\mathbf{x}_{t-1}, Q)/\alpha_t)p_t^Q(\mathbf{x}_{t-1}|\mathbf{x}_t) \\
&= \exp\left(\frac{[R_t(\mathbf{x}_t, Q) + A(\mathbf{x}_{t-1} - \mathbf{x}_t) - b]}{\alpha_t}\right) p_t^Q(\mathbf{x}_{t-1}|\mathbf{x}_t) \\
&\propto \exp\left(\frac{A\mathbf{x}_{t-1}}{\alpha_t} + \frac{-\|\mathbf{x}_{t-1}\|^2 + 2\langle\mathbf{x}_{t-1}, \mu_t^Q(\mathbf{x}_t)\rangle}{2\sigma_t^2}\right) \\
&= \exp\left(\frac{-\|\mathbf{x}_{t-1}\|^2 + 2\langle\mathbf{x}_{t-1}, \mu_t^Q(\mathbf{x}_t) + \sigma_t^2 A/\alpha_t\rangle}{2\sigma_t^2}\right).
\end{aligned}
$$

We conclude that

$$p_t^c(\cdot|\mathbf{x}_t) = \mathcal{N}(\mu_t^c, \sigma_t^2 I),$$

where

$$\mu_t^c = \sum_{i=1}^n \mu_t^i(\mathbf{x}_t) \odot Q_i + \frac{\sigma_t^2}{\alpha_t}\nabla_{\mathbf{x}_t} R_t(\mathbf{x}_t, Q).$$

Next, assume that the fictitious player was used. Repeating the same argument as above, we obtain

$$\mu_t^c = \sum_{i=1}^{n+1} \mu_t^i(\mathbf{x}_t) \odot Q_i + \frac{\sigma_t^2}{\alpha_t}\nabla_{\mathbf{x}_t} R_t(\mathbf{x}_t, Q).$$

Recall that $p^{n+1}(\mathbf{x}_{t-1}|\mathbf{x}_t)$ is defined as

$$p^{n+1}(\mathbf{x}_{t-1}|\mathbf{x}_t) \propto \prod_{i=1}^n p^i(\mathbf{x}_{t-1}|\mathbf{x}_t)^{1/n},$$

and thus,

$$\mu_t^{n+1}(\mathbf{x}_t) = \frac{1}{n}\sum_{i=1}^n \mu_t^i(\mathbf{x}_t).$$

Hence, we have

$$\mu_t^c = \sum_{i=1}^n \mu_t^i(\mathbf{x}_t) \odot \left(Q_i + \frac{1}{n}Q_{n+1}\right) + \frac{\sigma_t^2}{\alpha_t}\nabla_{\mathbf{x}_t} R_t(\mathbf{x}_t, Q).$$

$\square$

*Proof of Theorem 1.* Substituting definition of the efficiency functional $\mathcal{G}$, equation 4, in the fair division game in equation 1, we obtain the following optimization problem

$$Q_t = \arg\max_{Q\in\mathbb{Q}_t(\mathbf{x}_t)} \mathbb{E}_{\mathbf{M}\sim Q}\left[\sum_{i=1}^n\sum_{j=1}^m \mathbf{M}_{i,j}g_{ij}\right] - \beta_t D_{\mathrm{KL}}(Q||Q_{t+1}), \tag{10}$$

where $g_{ij} = \log u_{ij}(\hat{\mathbf{x}}_{t-1}(p), t-1)$ and

$$\mathbb{Q}_t = \left\{ Q \in \Delta(\mathbb{M}_{n,m}) : \mathbb{E}_{\mathbf{M} \sim Q} \sum_{i=1}^{n} \sum_{j=1}^{m} \mathbf{M}_{i,j} \phi_{ij} \preceq \mathbf{b} \right\}.$$

By the proof of Lemma 1, the problem in equation 10 is equivalent to

$$Q_t = \arg\min_{Q \in \mathbb{Q}_t(\mathbf{x}_t)} D_{\mathrm{KL}}(Q \| Q^*), \tag{11}$$

where

$$Q^*(M) \propto \exp\left( \sum_{i=1}^{n} \sum_{j=1}^{m} \mathbf{M}_{i,j} g_{ij} / \beta \right) Q_{t+1}(M).$$

$\square$

Assuming that $Q_{t+1}$ is decomposable with weights $Q_{ij}^{t+1}$, we find that

$$Q^*(M) \propto \prod_{i=1}^{n} \prod_{j=1}^{m} (e^{g_{ij}/\beta} Q_{ij}^{t+1})^{\mathbf{M}_{i,j}},$$

and thus, the allocation $Q^*(M)$ is decomposable with weights

$$Q_{ij}^* = \frac{e^{g_{ij}/\beta} Q_{ij}^{t+1}}{\sum_{i=1}^{n} e^{g_{ij}/\beta} Q_{ij}^{t+1}}.$$

We will solve the primal optimization problem in equation 11 in its dual form. Let $\phi(\mathbf{M}) = \sum_{i=1}^{n} \sum_{j=1}^{m} \mathbf{M}_{i,j} \phi_{ij}$. Assuming the set $\mathbb{Q}(\mathbf{x}_t)$ is non-empty (which is always the case for fairness constraints), the corresponding Lagrangian is

$$\max_{\lambda \geq 0, \gamma} \min_{Q} \mathcal{L}(Q, \lambda, \gamma),$$

where

$$\mathcal{L}(Q, \lambda, \gamma) = D_{\mathrm{KL}}(Q \| Q^*) + \lambda \left( \mathbb{E}_{\mathbf{M} \sim Q} \phi(\mathbf{M}) - \mathbf{b} \right) + \gamma \left( \sum_{\mathbf{M} \in \mathbb{M}_{n,m}} Q(\mathbf{M}) - 1 \right).$$

Taking derivative with respect to $Q(\mathbf{M})$ we obtain

$$\frac{\partial \mathcal{L}(Q(\mathbf{M}), \lambda, \gamma)}{\partial Q(\mathbf{M})} = \log Q(\mathbf{M}) + 1 - \log Q^*(\mathbf{M}) + \lambda \phi(\mathbf{M}) + \gamma = 0,$$

and thus,

$$Q(\mathbf{M}) = \frac{Q^*(\mathbf{M}) \exp(-\lambda \phi(\mathbf{M}))}{\exp(\gamma + 1)}.$$

We can now plug it into the Lagrangian and take derivative with respect to $\gamma$:

$$\frac{\partial \mathcal{L}(Q(\mathbf{M}), \lambda, \gamma)}{\partial \gamma} = \sum_{\mathbf{M}} \frac{Q^*(\mathbf{M}) \exp(-\lambda \phi(\mathbf{M}))}{\exp(\gamma + 1)} - 1 = 0.$$

Hence, we have

$$Q(\mathbf{M}) = \frac{Q^*(\mathbf{M}) \exp(-\lambda \phi(M))}{Z_\lambda}$$

with $Z_\lambda = \sum_{\mathbf{M}} Q^*(\mathbf{M}) \exp(-\lambda \phi(\mathbf{M}))$ and the dual problem reads

$$\max_{\lambda \geq 0} -\log(Z_\lambda) - \langle \mathbf{b}, \lambda \rangle.$$

Let $\lambda^*$ be a solution to the dual problem. The optimal allocation is expressed as

$$Q_t(\mathbf{M}) = \frac{Q^*(\mathbf{M})\exp(-\lambda^*\phi(\mathbf{M}))}{Z_{\lambda^*}} \propto \prod_{i=1}^{n}\prod_{j=1}^{m}(Q_{ij}^* e^{-\langle\lambda^*,\phi_{ij}\rangle})^{\mathbf{M}_{i,j}}.$$

Hence, it is also decomposable with weights

$$Q_{ij} = \frac{e^{g_{ij}/\beta}e^{-\langle\lambda^*,\phi_{ij}\rangle}Q_{ij}^{t+1}}{Z_j(\lambda^*)}$$

where

$$Z_j(\lambda^*) = \sum_{i=1}^{n}\exp\left(-\langle\lambda^*,\phi_{ij}\rangle\right)\exp(g_{ij}/\beta)Q_{ij}^{t+1}.$$

We conclude the proof by noticing that $Z_{\lambda^*} = \prod_{j=1}^{m}Z_j(\lambda^*)$.

### A.3 EFFECT OF GUIDANCE

In this section, we present additional experiments analyzing how guidance within the alignment step affects both performance metrics and computational cost. In particular, we study how results change when the parameter $\alpha$ is set to $\infty$ after the first $\tau$ iterations of the generative process.

We use the same experimental setup as in Section 4.2 with $n = 2$ players. All experiments are run on a single AWS EC2 G6e instance with 8 vCPU, 64 GB of memory, and a single 48 GB GPU. Alongside performance metrics, we report the average wall-clock time (in seconds) required to generate a batch of four images. The results are summarized in Table 4. Experiments involving fair allocations use envy-free constraints.

Notably, even with $\tau = 0$ guidance steps, our method consistently outperforms the baselines. Increasing the number of guidance steps yields further improvements, though at the cost of higher computation time. Interestingly, the computational overhead of projecting onto the fair allocation set is substantially larger when no guidance is used. This occurs because explicitly steering the models toward a fair and efficient division encourages better separation of interests, often leading to subsequent allocations already being fair. Without guidance, we find that fairness often needs to be imposed at every step during generation.

| Allocation | GenEval ↑ | | CLIP ↑ | | ImageReward ↑ | | VQA ↑ | | Time |
|---|---|---|---|---|---|---|---|---|---|
| | %images | %prompt | joint | min | joint | min | joint | min | |
| Implied | 54.85% | 84.69% | 27.06 | 18.76 | 0.31 | -1.14 | 0.758 | 0.684 | 10.09 |
| Uniform | 31.63% | 63.27% | 26.43 | 18.83 | -0.40 | -1.40 | 0.744 | 0.639 | 14.15 |
| Efficient, $\tau = 0$ | 61.99% | 94.90% | 28.24 | 20.21 | 0.45 | -0.91 | 0.867 | 0.791 | 25.2 |
| Efficient+Fair, $\tau = 0$ | 63.27% | 93.88% | 28.40 | 20.40 | 0.54 | -0.85 | 0.899 | 0.823 | 55.7 |
| Efficient, $\tau = 15$ | 79.59% | 96.94% | 29.34 | 21.02 | 1.05 | -0.51 | 0.937 | 0.894 | 32.9 |
| Efficient+Fair, $\tau = 15$ | 81.63% | 97.96% | 29.37 | 21.01 | 1.06 | -0.51 | 0.948 | 0.909 | 46.2 |
| Efficient, $\tau = 50$ | 85.97% | 98.98% | 29.60 | 21.17 | 1.12 | -0.49 | 0.958 | 0.924 | 49.6 |
| Efficient+Fair, $\tau = 50$ | 85.71% | 98.98% | 29.64 | 21.23 | 1.13 | -0.47 | 0.960 | 0.925 | 64.1 |

Table 4: Performance of Divide-and-Denoise on coordinating 2 models conditioned on different concepts under varying numbers of guidance steps $\tau$.

### A.4 CONFLICTING PROMPTS DATASET

We constructed a custom dataset comprising 40 examples, each designed with conflicts between individual prompts. Specifically, the first 30 prompts involve either object + semantics or semantics + semantics compositions: 10 focus on conflicting attributes (e.g., "an image with a blue lake" and "an image with violet trees"), while the remaining 20 capture conflicting semantic combinations (e.g., "an image with a desert" and "an image with a snowy mountain"). The last 10 prompts are of the form semantics + semantics + object, where at least one pairing is conflicting.

We list the prompts below:

an image with a blue lake, an image with violet trees

an image with a green car, an image with a pink forest

an image with a yellow elephant, an image with a grey desert

an image with a rainbow-colored dog, an image with a black-and-white city

an image with a brown flamingo, an image with a purple swamp

an image with a transparent car, an image with a glowing forest

an image with a golden cloud, an image with a black ocean

an image with a white bus, an image with a bright orange snowfield

an image with a blue cat, an image with a black sofa

an image with a red eagle, an image with a grey sky

an image with a desert, an image with a snowy mountain

an image with a jungle, an image with an icy glacier

an image with a burning forest, an image with a frozen river

an image with a tropical beach, an image with a volcanic eruption

an image with a futuristic city, an image with a medieval castle

an image with a stormy sky, an image with a calm lake

an image with a carnival, an image with a haunted graveyard

an image with an underwater city, an image with a floating island

an image with a sunny meadow, an image with a meteor shower

an image with a winter tundra, an image with a blooming spring forest

an image with snowy mountains, an image with a camel

an image with a rainforest, an image with a penguin

an image with a rocky cliffside, an image with a telephone booth

an image with an iceberg, an image with a windmill

an image with a busy highway, an image with a deer

an image with a street, an image with a jaguar

an image with a blizzard, an image with a giraffe

an image with a desert oasis, an image with a moose

an image with a rice field, an image with a Ferris wheel

an image with a savanna, an image with a skyscraper

an image with a desert canyon, an image with snowy peaks, an image with a tropical parrot

an image with a modern city skyline, an image with a rural farm, an image with a horse

an image with a sunflower field, an image with snowy mountains, an image with a polar bear

an image with a grassy soccer field, an image with volcanic ash clouds, an image with a motorcycle

an image with a frozen lake, an image with a tropical beach, an image with a palm tree

an image with an iceberg, an image with stormy skies, an image with a cow

an image with a grassy meadow, an image with a tropical sun, an image with a snowman

an image with a sandy beach, an image with the aurora borealis, an image with an elephant

an image with a wheat field, an image with a futuristic glass dome city, an image with a steam train

an image with a rocky cliffside, an image with a rainbow sky, an image with a boat

## A.5 ADDITIONAL QUALITATIVE RESULTS

We provide additional qualitative comparisons of Divide-and-Denoise with Implied Allocation and Uniform Allocation baselines. The experimental setup is described in Section 4. Additionally, we show the performance of Divide-and-Denoise without allocation guidance, i.e., $\alpha \to \infty$. This variant is faster since it does not require backpropagation for the coordinated update. While this version already improves image quality over the uniform baseline, allocation guidance is important to ensure that each model reliably denoises its assigned region.

The images are shown in Figures 4, 5, and 6. Each row corresponds to one coordination mechanism: Stable Diffusion (implied allocation baseline), Composable Diffusion (uniform allocation baseline), and Divide-and-Denoise without ($\alpha \to \infty$) and with ($\alpha = 1$) guidance. Each column corresponds to a fixed set of concepts. For each combination of method and concept set, we generate a batch of 4 images.

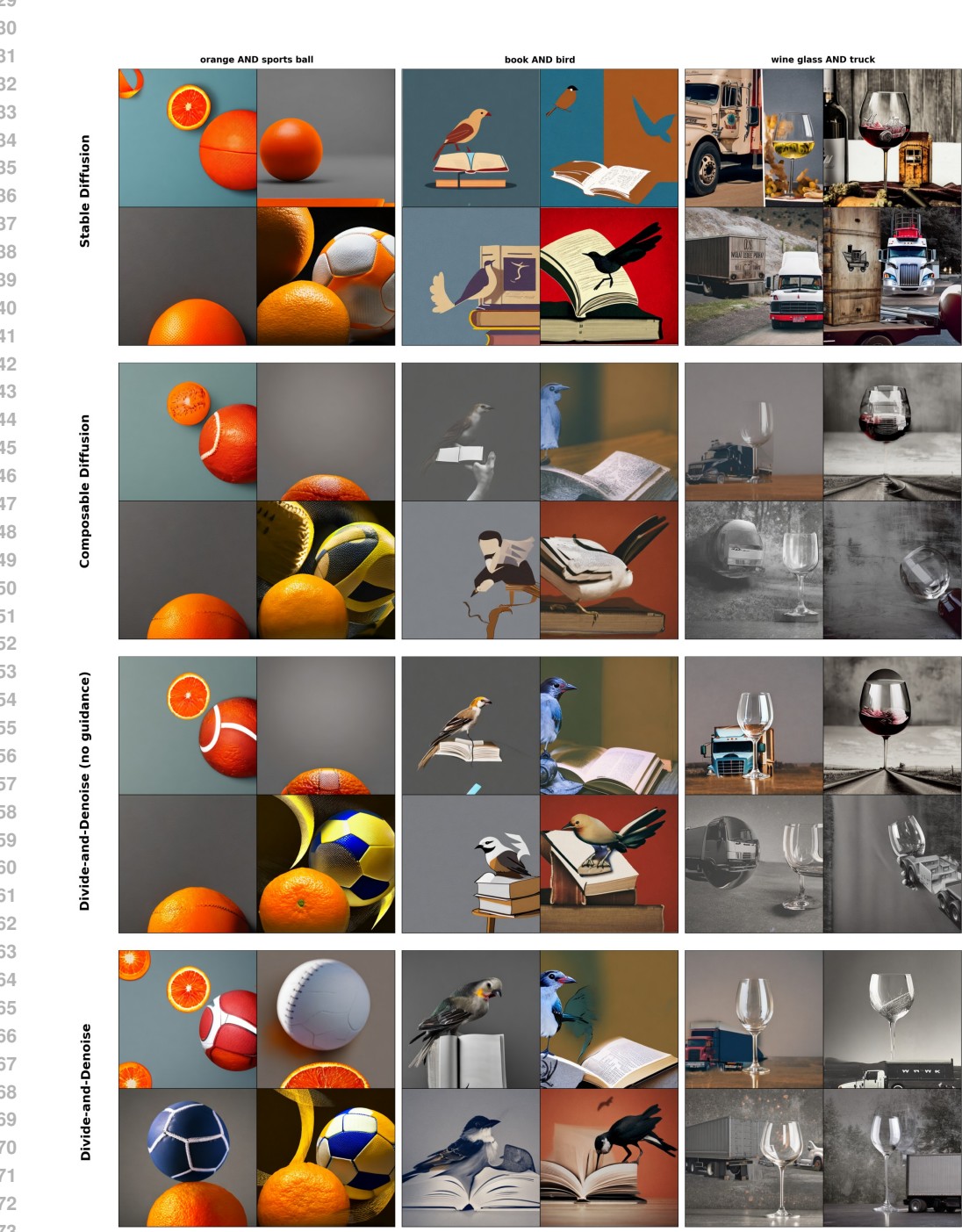

Figure 4: Qualitative comparison on GenEval benchmark (2 players)

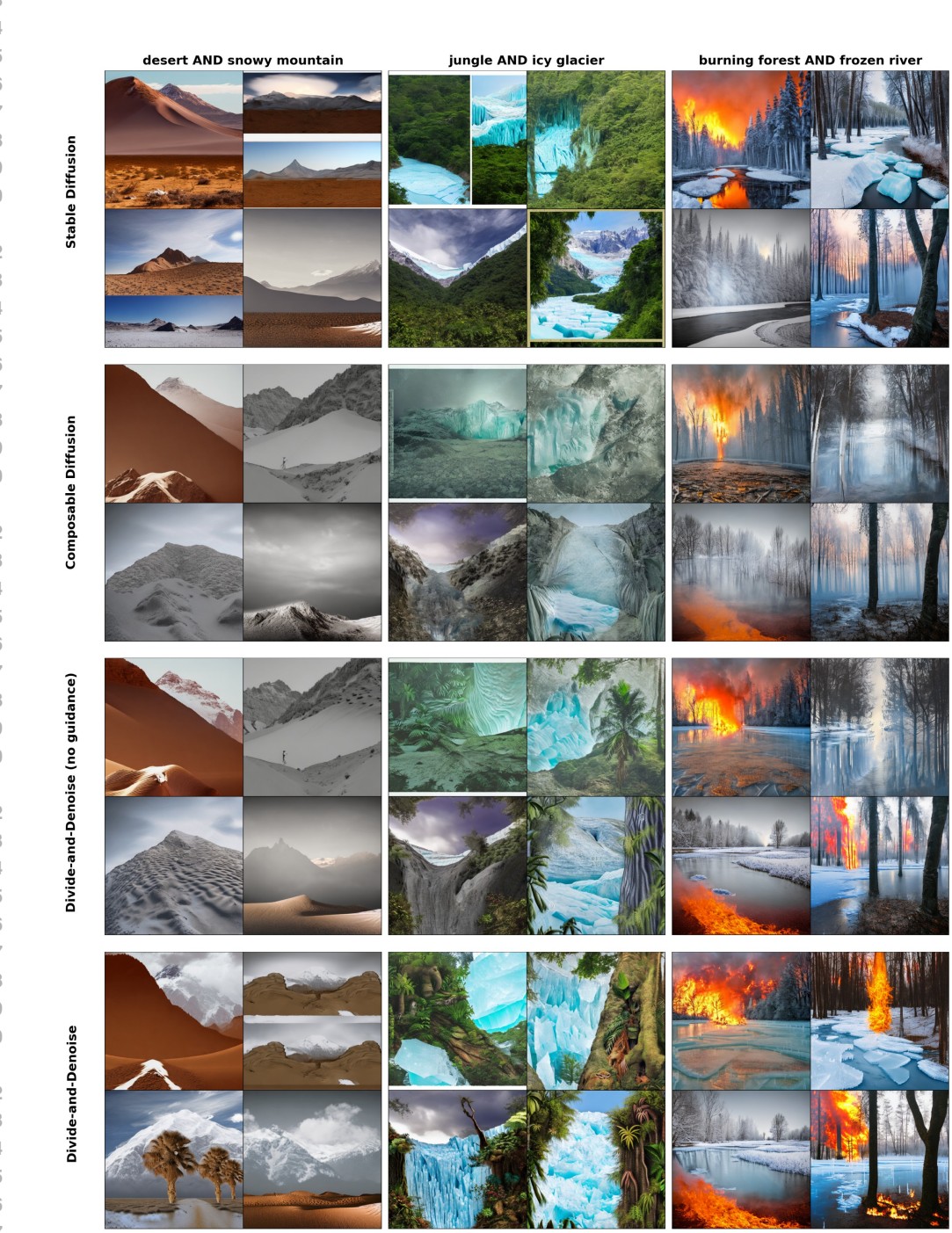

Figure 5: Qualitative comparison on Conflict dataset (2 players)

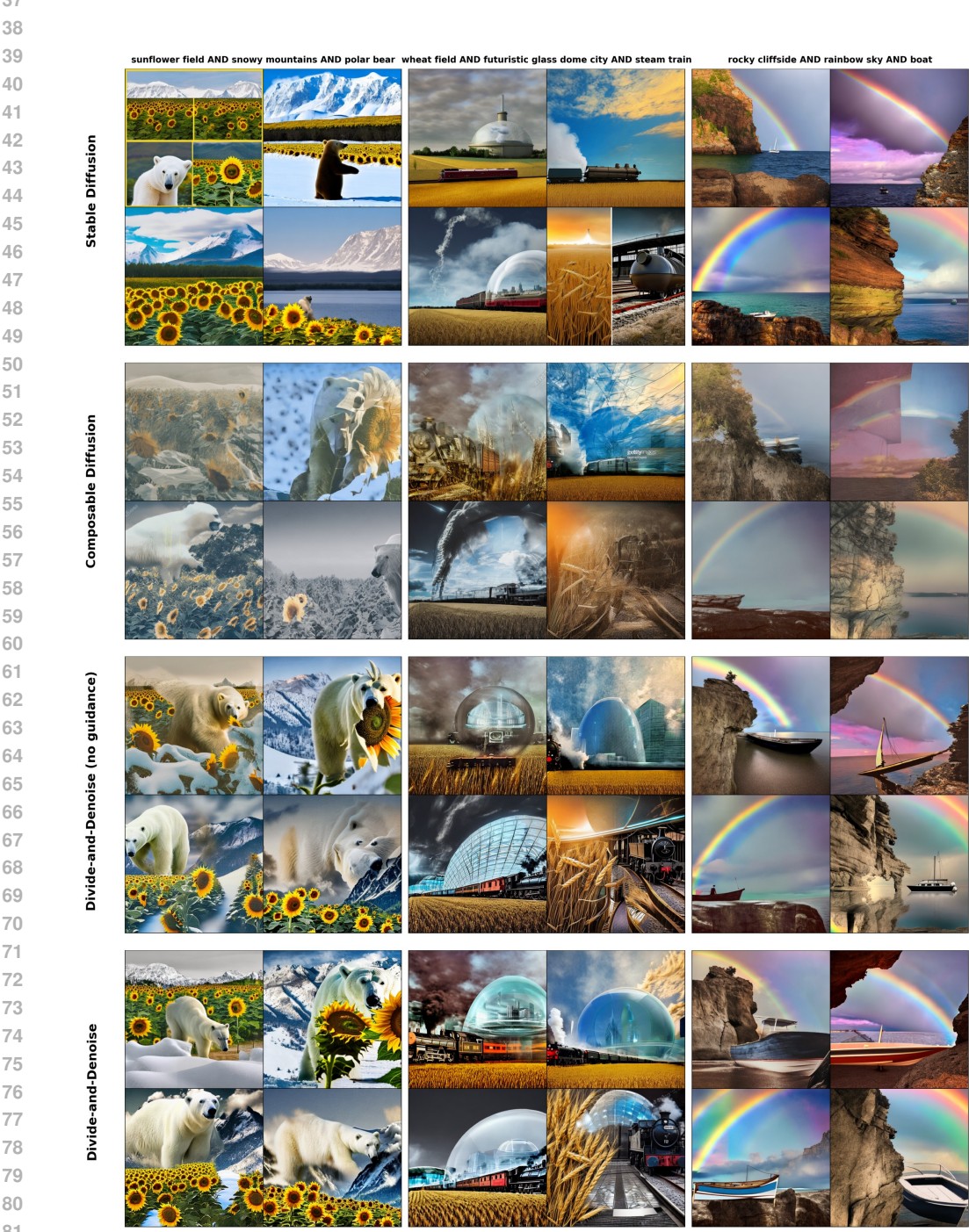

Figure 6: Qualitative comparison on Conflict dataset (3 players)

