# OpenReview forum: "DIVIDE-AND-DENOISE: A GAME THEORETIC METHOD FOR FAIRLY COMPOSING DIFFUSION MODELS"
_ICLR.cc/2026/Conference — Submitted to ICLR 2026_

### Official Review · Reviewer_9aeu · 2025-10-30

**Soundness:** 2
**Presentation:** 3
**Contribution:** 2
**Rating:** 4
**Confidence:** 3

**Summary:**

The paper proposes a game-theoretic approach to combine multiple different text-to-image diffusion models. A crucial constraint is that the models must have the same latent dimensions. Ensuring fairness when dividing the noise maps among the models prevents the collapse into a single concept and only generating this concept. The experimental results show that the approach doesn't suffer from collapse to a single concept and can generate all the objects present in the prompt using different models.

**Strengths:**

- The idea of fusing different models to generate an image is very intriguing.
- The paper is well written, and even the mathematical details are easy to understand.

**Weaknesses:**

- The figures in the paper have low resolutions. The text in the images is not readable.
- A few more example images would be nice to illustrate what makes this approach better than other approaches.
- The evaluation is not very thorough. For example for the generation of multiple objects and the attribute allocation only figure 3 is shown as evidence.
- It is not clear why the prompts used in Section 4.3 are out-of-distribution.

Minor:
- In line 404, 412 and 413 the citations seem to be missing.
- The figure number in line 423 is not correct
- In line 466 the table number is not correct

**Questions:**

Q1: I might have missed it, but how is the fairness ensured when dividing the pixels to the models?
Q2: Why do the pixels have to be distributed to a fixed model? Wouldn't it also be possible, especially when two areas overlap, to average over the noise maps of multiple models?
Q3: How does VQA measure the compositional correctness? If I am not mistaken, an image can be composed in different ways, while the VQA can still be correct.
Q4: Why are there values missing for VQA in table 1?
Q5: Why are the prompts used in Section 4.3 OOD? What does it mean if there are "conflicts between individual prompts"?

**Details Of Ethics Concerns:**

There are no concerns.

---

> ### Author Response · Authors · 2025-11-25
>
> Thank you for your thoughtful and constructive review. We feel very encouraged that you found the core idea "very intriguing" and the paper, including the mathematical details, was "well written" and "easy to understand."
>
> We will address your specific questions, many of which highlight presentation flaws in our original submission that we have now corrected.
>
> ### 1. On Presentation (Weaknesses & Minor Issues)
> We sincerely apologize for the poor presentation quality of the submitted draft. You are entirely correct that the low-resolution figures, missing citations, and incorrect figure/table references are unacceptable.
> * Fixed: All figures (Figures 1, 2, and 3) should now be visible, and all text within them is now sharp and readable.
> * Fixed: We have corrected all placeholder citations (e.g., [?]) and formatting errors.
> * Fixed: We have corrected the erroneous figure/table references. The text in the results sections (now 4.2, 4.3, 4.4) now correctly points to Table 1, Table 2, and Table 3, respectively.
> * More Examples: We agree that more qualitative examples strengthen the paper. While balancing page limits, we have ensured the existing figures are clear and have added further qualitative results to the Appendix.
>
> ### 2. Responses to Questions
> > Q1: How is fairness ensured when dividing the pixels?
> >
> > This is an excellent question that gets to the core of our method. Fairness is not an emergent property but is explicitly enforced as a hard constraint in the first optimization problem (equation (1)).
> >
> > As defined in Section 3.2, the allocation $Q_t$ must be chosen from the set of "fair" allocations. This set is defined by a series of linear constraints in Equation 5. We have added  Example 1, showing how the coefficients $\phi_{ij}$ encoding these constraints can be defined to match standard game-theoretic fairness notions like envy-freeness, proportionality, and equitability. Our solver must then find an efficient allocation that satisfies these hard fairness constraints. In practice, we use the cvxpy library to solve the dual problem formulated in Theorem 1. Note that dual formulation has only $l$ variables, where $l$ is the number of linear constraints.
>
> > Q2: Why must pixels be distributed to a fixed model? Why not average overlapping regions?
> It is a delicate but important point. On the one hand, as we note in the introduction and show in our experiments, a simple averaging approach over the entire image frequently fails, blending two concepts together and causing neither to be represented clearly. Moreover, we noticed that overly smooth allocations provide a weak signal for the allocation guidance. That motivates us to define reward components on binary assignment matrices $M$. At the same time, we agree that regions of responsibility can naturally overlap, for example, along object boundaries. Our algorithm accounts for this in two ways. First, rather than choosing a single binary assignment $M$, we optimize for the fair allocation $Q$, a distribution over all possible assignments. As discussed in Section 2.3, our allocations are closely related to fractional assignments, allowing soft division of responsibility where appropriate. In principle, nothing forces the allocation to be non-uniform if such division appears to be efficient. In practice, however, the efficient allocations we observe tend to be nearly degenerate. For this reason, we proposed a fictitious player that enforces averaging of pixels in the overlapping regions. For regions where no single concept-model has a strong claim, our fictitious (n+1)-st player takes over. Notice how the pixels allocated to the fictitious player are uniformly distributed across real players. We argue that it is a more principled way to handle “overlap” than simply averaging the conflicting proposals.
>
> > Q3: How does VQA measure compositional correctness?
> > Thank you for your question. We have provided a more detailed description of the evaluation metrics in Section 4.1. Similar to the other metrics, we use the VQA score to assess how well the image represents single- or multi-concept prompt $y$. The VQA score uses a visual-question-answering model to produce an alignment score by computing the probability of a "Yes" answer to a simple "Does this figure show '{y}'?" question.
>
> > Q4: Why are there values missing for VQA in Table 1?
> >
> > This was a formatting error in the submitted draft, and we apologize. The VQA results for the "Uniform" 3-object baseline were indeed missing. We have corrected this in the revised Table 1.

---

> > ### Author Response · Authors · 2025-11-25
> >
> > > Q5: Why are the prompts used in Section 4.4 (old 4.3) OOD? What does "conflicts" mean?
> > >
> > > This is an excellent point, and you are right that "OOD" was a confusing and imprecise term. Another reviewer made the same valuable suggestion.
> > >
> > > We have renamed this section to "4.4 Concepts with Conflict" in the revised paper. These prompts are "conflicting" because they combine concepts that are semantically or visually unnatural to see together (e.g., "an image with a desert" and "an image with a snowy mountain"). This test is designed to be extremely challenging for a single multi-concept or joint prompted model, which is typically trained on data where concepts have natural co-occurrences. Our method, by coordinating separate expert models, is able to robustly generate these novel combinations.
> >
> > We thank you again for your questions and constructive feedback. Your questions have helped us identify and fix key presentation flaws and clarify the paper's core contributions. We hope the revised manuscript is now much stronger as a result.

---

### Official Review · Reviewer_roNH · 2025-10-31

**Soundness:** 3
**Presentation:** 2
**Contribution:** 3
**Rating:** 4
**Confidence:** 4

**Summary:**

The paper “Divide-and-Denoise” proposes a game-theoretic framework for compositional sampling from multiple pre-trained diffusion models. Rather than directly averaging denoising predictions (as in MultiDiffusion or joint-prompt methods), the authors formulate the problem as a fair division game, where latent coordinates are “goods” and each diffusion model acts as a “player.” This elegant formulation allows the model to dynamically allocate spatial responsibility among different diffusion processes in a principled, temporally coherent, and fairness-aware way.

The method alternates between two tightly coupled updates at each diffusion step:
	1.	Compositional denoising, which generates a latent proposal based on soft region assignments Q_t;
	2.	Dynamic allocation, which optimizes Q_t via a bilevel optimization that enforces fairness, smoothness, and attention alignment across time.

A key novelty is the introduction of the alignment score derived from cross-attention maps, which measures semantic consistency between denoised regions and textual prompts. The paper also introduces a “fictitious player” to handle unassigned or background regions, ensuring that all latent coordinates are properly modeled. Theoretical analysis leads to a closed-form softmax-like solution for Q_t (Theorem 2), while alternating optimization jointly refines both the denoising kernel and spatial allocation.

**Strengths:**

(1) Conceptual originality: The use of game theory and fair division in diffusion model coordination is highly innovative and goes beyond heuristic compositional fusion.

(2) Theoretical rigor: The bilevel formulation, connection to entropy-regularized MDPs, and derivations (Theorems 1–2) are mathematically sound and clearly motivated.

(3) Strong empirical performance: On multi-object and attribute-binding tasks, Divide-and-Denoise significantly reduces object overlap and color confusion, outperforming joint-prompt and MultiDiffusion baselines.

**Weaknesses:**

(1) Computational overhead: Alternating updates for Q_t and p_t^c introduce nontrivial cost during inference.

(2) Dependence on cross-attention quality: The allocation accuracy relies heavily on stable and interpretable attention maps.

(3) Limited evaluation scope: Current experiments are restricted to text-to-image synthesis; demonstrating broader modality coverage would further strengthen the claim of generality.

**Important** (4 )Many figures are blurry, making them nearly unreadable. Several references are missing or incorrectly formatted, which severely reduces the paper’s professionalism and readability.

**Questions:**

None

---

> ### Author Response · Authors · 2025-11-25
>
> We are deeply grateful to you for your positive, thorough, and insightful review. We are particularly pleased that you recognized the conceptual originality of our game-theoretic formulation and its theoretical rigor, including the bilevel optimization, the role of the fictitious player, and the alignment score. Your summary accurately captures the core of our contribution.
>
> We will address the weaknesses you identified, starting with the most critical one.
>
> ### 1. On Presentation Quality (Weakness 4)
>
> You are entirely correct. The state of the submitted manuscript was unacceptable, and we sincerely apologize for the blurry figures, missing citations, and formatting errors. These presentation issues were our top priority for this revision.
>
> * Fixed: We have re-rendered all figures (including Figure 1, 2, and 3) at high resolution, and they are now clear and legible.
> * Fixed: We have meticulously corrected all placeholder citations and formatting errors throughout the paper.
> * Fixed: The paper has been thoroughly edited and reformatted to adhere to all ICLR guidelines, including the page limit.
>
> We thank you for evaluating the paper's scientific merit despite these significant presentation flaws.
>
> ### 2. On Computational Overhead (Weakness 1)
>
> This is a valid point. Our method intentionally trades a measure of computational cost for a more robust and fair coordination strategy. The sequential updates are indeed more expensive than a single forward pass.
>
> We have sought to balance this trade-off in our final algorithm. As noted at the end of Section 3, we employ a "one-step" warm-start approximation, which provides a stable lookahead for the game without requiring full convergence at each timestep. We believe this is a reasonable compromise for the significant gains in performance.
>
> We plan to further address computational concerns by including additional ablation studies in the final version of the manuscript. We are currently running experiments that reduce the number of steps in which allocation guidance is computed, since backpropagation brings the primary computational overhead relative to the baselines.
> Our preliminary results on the 2-concept experiment indicate that even without allocation guidance (i.e., setting $\alpha \rightarrow \infty$), our algorithm still reliably outperforms both baselines while significantly lowering computational cost. We expect, however, that allocation guidance will play an increasingly important role as the number of players grows.
>
> ### 3. On Dependence on Cross-Attention (Weakness 2)
>
> This is an excellent point and a key area for future work. Our framework's performance is indeed coupled to the quality of the utility functions, which we derive from cross-attention maps in the text-to-image domain.
>
> While we found these maps to be surprisingly effective, our framework is modular. As we state in our conclusion, one of the most exciting future directions is to explore alternative utility definitions, such as learned utility models or different interpretable signals, which could be drop-in replacements for attention maps. This is especially relevant for extending the method to domains where attention is less interpretable or unavailable.
>
> ### 4. On Limited Evaluation Scope (Weakness 3)
>
> We chose text-to-image synthesis as our primary testbed because it provides a well-understood, verifiable, and visually intuitive domain to develop and validate our coordination framework. Since ground-truth is not available, we felt that GenEval could be a strong way to evaluate the performance of our proposed coordination strategy.
>
> However, as you correctly note, the core mechanism is domain-general. We strongly agree that demonstrating this generality is the most important next step. We have explicitly highlighted the extension to other modalities (e.g., text-to-audio, text-to-graph, or multi-modal synthesis) as the primary direction for future work in our conclusion.
>
> We thank you again for your constructive feedback and your supportive assessment. We believe the revised manuscript, with its significantly improved presentation and clarifications, is now a much stronger paper that addresses your concerns.

---

### Official Review · Reviewer_nmYV · 2025-10-31

**Soundness:** 2
**Presentation:** 1
**Contribution:** 2
**Rating:** 0
**Confidence:** 3

**Summary:**

This paper aims to tackle compositional generation with diffusion models by introducing "Divide-and-Denoise", a game theoretic sampling procedure that composes multiple pretrained diffusion model "player" models via fair division of the latent space at every denoising step. The method alternates between (i) an allocation step that infers soft segmentations by solving a fairness-constrained optimization using utilities derived from cross-attention maps, and (ii) a denoising step whose optimal Gaussian kernel has a mean that combines per-model updates masked by the allocation plus a guidance term driven by an alignment score; a fictitious background player and a KL term encourage sensible coverage and temporal smoothness.

**Strengths:**

- ***Interesting & principled idea***: Recasts compositional generation as a fair-division game over soft region allocations, using cross-attention.

**Weaknesses:**

- ***Writing quality***: The paper appears incompletely prepared at submission time. In the experiments section there are placeholder “?” citations, tables that overflow horizontally, and tables with missing entries. The manuscript also exceeds the 9-page limit, suggesting the writing and formatting were not finalized. These presentation issues significantly hinder readability and raise concerns about diligence in preparing the submission.

- ***Experimental setups***: The Joint Prompt setup appears to be an extremely weak baseline. With such a simple enumeration-style prompt, the model has a high probability of failure. Instead, the authors should compare results when using a language model to generate natural prompts containing multiple objects. In the same vein, averaging is also far too simple as a baseline. It seems strange to expect that averaging score values from different conditions would work well.

- ***Prompt division***: This paper focuses on effectively dividing and combining generation from multiple players, yet it doesn't address how to divide the conditions among them. For example, if there's a long prompt, there is a need to determine how to distribute its contents to each player. With the current approach, I have serious doubts about whether this can work for scenarios with complex multiple relations.

- ***The title of Section 4.3***: I don't understand why this is considered out-of-distribution at all. Wouldn't "conflict prompt" be more appropriate?

**Questions:**

N/A

---

> ### Author Response · Authors · 2025-11-25
>
> We thank the reviewer for their time and detailed feedback. We will address the primary concerns regarding the paper's presentation, the experimental baselines, the method's scope, and the terminology we used.
> ### 1. On Presentation Quality and Completeness
> We apologize for the presentation issues in the submission. These are easily corrected and, in our revised manuscript,
> * All placeholder citations (`?`) have been addressed
> * All tables have been reformatted to fit within the page margins and are now clear and legible.
> * The paper has been edited for clarity, brevity, and correctness. It now strictly adheres to the page limit.
> * All figures should now be clearly visible.
> We thank the reviewer for their patience in evaluating the work despite the initial issues.
> ### 2. On Experimental Setups and Baselines
> This appears to be the central misunderstanding, and we thank the reviewer for the opportunity to clarify our paper's core contribution and the rationale for our baselines.
> Aim of work: The main purpose of our work is not to improve image generating conditioned on a complex prompt by decomposing it into simpler subprompts. Instead, we are solving a multi-model coordination problem. Given a pre-selected team of $n$ specialist models (players), unknown in their capabilities, we aim for the best inference-time strategy to combine them. In particular, our method does not assume access to any “multi-concept” model that combines expertise of individual players. Instead, we seek an inference-time solution that avoids expensive and often infeasible training of such a model. Our work is therefore situated within the growing field of composition of generative models, including works like Yilun Du et al. Reduce, Reuse, Recycle: Compositional Generation with Energy-Based Diffusion Models and MCMC and Marta Skreta et al. The Superposition of Diffusion Models Using the Itô Density Estimator, which also seeks to combine specialist models at inference time.
>
> Justification of Baselines: Our baselines were chosen specifically to provide fair and direct comparisons for this coordination task.
> * Joint Prompt (Implied Allocation): The reviewer notes that a "simple enumeration-style prompt" is an "extremely weak baseline". We use this simple enumeration prompt because it is the fairest possible comparison: it uses the exact same limited information as available to our method (i.e., a list of $n$ distinct concepts, $y_1, ..., y_n$) without any explicitly given relational knowledge.
> * The "LLM Oracle": The reviewer suggests using an LLM to generate "natural prompts" as a stronger baseline. We believe this would be an unfair comparison. An LLM-generated prompt (e.g., "a dog sleeping next to a cat") acts as a powerful, domain-specific oracle. It injects rich, external knowledge about how concepts naturally relate. A team of specialized models would not typically have access to such information. This limitation is shared by all "blind" compositional approaches. We fully agree that if a powerful model trained on complex combinations of multiple concepts is available, then the best strategy is to leverage that model directly, possibly enhancing it with guidance techniques.
> Our method is intentionally application-agnostic and operates without any such oracle, assuming no prior knowledge of concept relationships. This is crucial for our goal of building a framework that generalizes to domains beyond text-to-image generation (e.g., robotics, protein design), where such a convenient "joint prompting" oracle does not exist. We have explicitly clarified this rationale in the revised Section 4.
> * Averaging (Uniform Allocation): The reviewer finds this "far too simple." We included this baseline because averaging the models' proposals (or scores) is the de facto standard and simplest method for multi-model coordination. It is a popular baseline and can indeed work well when models already have quite different preferences (e.g., each model inherently wishes to claim a different region). For example, the recent work of Arwen Bradley et al. “Mechanisms of Projective Composition of Diffusion Models” introduces a notion of projective composition and show that score averaging matches the corresponding pushforward of the composed distribution when the concepts are orthogonal in an appropriate feature space. In the image domain, this orthogonality often holds for object–background combinations, but can break down for compositions involving multiple objects.
> We have revised Section 4 to make this critical distinction between coordination and prompting much clearer.

---

> > ### Author Response · Authors · 2025-11-25
> >
> > ### 3. On the Scope ("Prompt Division")
> > The reviewer notes that our paper does not address how to automatically divide a long, complex prompt into a set of players. However, our contribution is the coordination mechanism itself, which operates on a fixed set of specialized models. Because we focus on inference-time coordination, we cannot assume access to arbitrary specialized models. A complementary system could indeed select, from a given pool, the subset of specialized models best suited for a task, which, for example, can correspond to prompt decomposition in the text-to-image setting.
> > ### 4. On the Title of Section 4.3 (now 4.4)
> > This is an excellent point. The reviewer is correct that "Conflicting Concepts" is a far more accurate and appropriate title than "Out-of-Distribution Generalization." We have renamed this section in the revised manuscript to "4.4 Concepts with Conflict".
> >
> > We thank the reviewer again for their constructive feedback, which has helped us significantly clarify the paper's core problem, contribution, and scope. We recognize that some of the misunderstanding stemmed from our original presentation, which placed too much emphasis on the particular text-to-image setting. In response, we have reorganized the manuscript to highlight our main objective and contribution: how to divide the workload among specialized models in a principled and general way.
> > We hope the revised manuscript, with its corrected presentation and clearer framing, will merit a new evaluation.

---

### Meta-Review · Area_Chair_swbt · 2026-01-06

**Summary:**

The paper proposes a framework for coordinating multiple diffusion models through alternating division and denoising steps. The reviewers raise concerns about writing quality, limited evaluation, method robustness, and additional computational overhead.

**Reviewer Concerns:**

Reviewer concerns addressed by the rebuttal:

1. The writing issues were partially addressed (Reviewers nmYV, 9aeu). Some minor formatting issues still remain (e.g., quotation marks such as ”xxx”).
2. Additional details and clarifications about the proposed method were provided (Reviewer 9aeu).

Reviewer concerns not fully addressed by the rebuttal:

1. Concerns about weak baselines and limited evaluation remain (Reviewers nmYV, roNH).
2. Concerns about method robustness remain, including dependence on cross-attention quality (Reviewers nmYV, roNH).
3. Concerns about computational overhead remain (Reviewer roNH).

**Reviewer Scores:**

If the reviewers participate fully in the discussion, Reviewer 9aeu may raise their score to 6, since most of their questions are addressed. However, Reviewers roNH and nmYV may still maintain their negative scores, as the rebuttal and revision do not provide sufficiently convincing evidence to resolve their concerns.

---

### Decision · Program_Chairs · 2026-01-26

Reject